# Application of Self-Assembled Polyarylether Substrate in Flexible Organic Light-Emitting Diodes

**DOI:** 10.3390/mi14050920

**Published:** 2023-04-24

**Authors:** Hsin-Yi Wen, Yu-Shien Lu, Cheng-Yan Guo, Mei-Ying Chang, Wen-Yao Huang, Tung-Li Hsieh

**Affiliations:** 1Department of Chemical and Materials Engineering, National Kaohsiung University of Science and Technology, Kaohsiung 80778, Taiwan; hsinyiwen@nkust.edu.tw; 2Department of Photonics, National Sun Yat-sen University, Kaohsiung 80424, Taiwan; a123s16zxc@gmail.com (Y.-S.L.); mychang01@mail.nsysu.edu.tw (M.-Y.C.); 3Department of Electronics Engineering, National Kaohsiung University of Science and Technology, Kaohsiung 80778, Taiwan; gcy626@gmail.com

**Keywords:** copolymer, flexible substrate, organic light emitting diode, fluorescent material, thermal stability

## Abstract

The structure used in this study is as follows: substrate/PMMA/ZnS/Ag/MoO3/NPB/Alq3/LiF/Al. Here, PMMA serves as the surface flattening layer, ZnS/Ag/MoO3 as the anode, NPB as the hole injection layer, Alq3 as the emitting layer, LiF as the electron injection layer, and aluminum as the cathode. The properties of the devices with different substrates were investigated using P4 and glass, developed in the laboratory, as well as commercially available PET. After film formation, P4 creates holes on the surface. The light field distribution of the device was calculated at wavelengths of 480 nm, 550 nm, and 620 nm using optical simulation. It was found that this microstructure contributes to light extraction. The maximum brightness, external quantum efficiency, and current efficiency of the device at a P4 thickness of 2.6 μm were 72,500 cd/m^2^, 1.69%, and 5.68 cd/A, respectively. However, the maximum brightness of the same structure with PET (130 μm) was 9500 cd/m^2^. The microstructure of the P4 substrate was found to contribute to the excellent device performance through analysis of the AFM surface morphology, film resistance, and optical simulation results. The holes formed by the P4 substrate were created solely by spin-coating the material and then placing it on a heating plate to dry, without any special processing. To confirm the reproducibility of the naturally formed holes, devices were fabricated again with three different emitting layer thicknesses. The maximum brightness, external quantum efficiency, and current efficiency of the device at an Alq3 thickness of 55 nm were 93,400 cd/m^2^, 1.7%, and 5.6 cd/A, respectively.

## 1. Introduction

The earliest organic light-emitting diode (OLED) can be traced back to 1963, when M. Pope et al. published a paper on OLED [1]. They used a single layer of anthracene crystal as the emitting layer, which exhibited a weak luminescence phenomenon. However, due to the lack of mature thin film technology at that time, the problems of high driving voltage and poor luminous efficiency could not be solved, so it was not taken seriously at the time. Subsequently, physical vapor deposition (PVD) was introduced for the deposition of organic polycrystalline thin films, which solved the problem of high driving voltage. However, the unevenness and instability of polycrystalline films still hindered their luminous efficiency [2,3]. However, in 1987 C.W. Tang and Vanslyke from Kodak in the United States published a paper on OLED using vacuum evaporation technology to deposit Alq3 and HTM-2 to form a heterostructure OLED device. The resulting device had a driving voltage of less than 10 V and an external quantum efficiency of up to 1% green light OLED, making the application of organic luminescent materials and devices feasible and valuable for research [4].

Polymer light-emitting diodes (PLEDs), which use spin coating technology to coat polyphenylene vinylene (PPV) as the emitting layer, and indium tin oxide (ITO) and aluminum as the anode and cathode respectively, were first developed in 1990 by J.H. Burroughes and his colleagues at the University of Cambridge in the UK. They fabricated a single-layer electroluminescent device ITO/PPV/Al [5]. Spin coating technology can complete the film coating in a normal atmospheric environment, making it simpler and more convenient than vacuum deposition technology. In order to improve its luminous efficiency, solubility, film-forming properties, and to modulate its emission wavelength, many derivatives of PPV have been developed, such as CN-PPV [6], RO-PPV [7], and PF [8]. In 1992, Gustafson and his colleagues at the UNIAX corporation fabricated flexible PLEDs by spin-coating MEH-PPV onto a conductive polymer—polyaniline (PANI)—and a transparent polyethylene terephthalate (PET) substrate, and depositing metal electrodes on top of it [9]. The molecular weight of the polymer materials is usually in the tens of thousands or even hundreds of thousands, while small molecule materials generally range from several hundred to several thousand. Therefore, the molecular weight of small molecule materials is easier to control than polymers, and it is easier to achieve the requirements for material yield and purity during purification and synthesis processes. Although controlling the molecular weight of small molecules is easier than that of polymers, their thermal stability and mechanical properties are worse than those of polymers. Therefore, PLED devices can withstand higher current density and higher temperature environments compared with OLED devices, making their applications more diverse.

The strengths of flexible organic light-emitting diodes (OLEDs) include their lightweight and bendable nature, which allows for easy integration into a variety of applications. OLEDs are known for their low power consumption and high energy efficiency compared with traditional lighting technologies. The response time of flexible OLEDs is significantly faster than other display technologies, reducing motion blur and providing a smoother viewing experience, while the organic materials used in OLEDs can be easily manipulated, allowing for customizable and scalable designs. However, flexible organic light-emitting diodes (OLEDs) also have weaknesses, including the fact that they typically have a shorter lifetime compared with other display technologies such as LCDs or LEDs. Organic materials in OLEDs are sensitive to moisture and oxygen, which can degrade their performance over time. The fabrication process for flexible OLEDs is complex and requires precise control over material deposition and encapsulation and the production costs for flexible OLEDs can be higher than other display technologies, especially for large-scale applications.

To overcome the problem of current injection, the chemical structures of each layer in OLED are designed for different functions, and various materials are synthesized to provide injection of electrons and holes, assist with the transmission of electrons and holes, or block their transmission, etc. The OLED device includes an anode, hole injection layer (HIL), hole transport layer (HTL), emitting layer (EMT), electron transport layer (ETL), electron injection layer (EIL), and cathode. Typically, OLED devices that are efficient and have a long lifespan use a multi-functional layer device structure. The structure of the device in this experiment is substrate/(PMMA)/ZnS/Ag/MoO_3_/NPB/Alq3/LiF/Al. The device was fabricated to investigate the characteristics of the devices using commercially available flexible substrates and polymer materials developed in our laboratory for flexible OLEDs, as well as the characteristics of the devices after adding PMMA, which is commonly used for surface flatness [10,11].

## 2. Materials and Methods

### 2.1. Preparation of P4 and PMMA Materials

P4 is an alternating copolymer developed in our laboratory that exhibits good thermal stability. The synthesis of alternating copolymers involves preparing two monomers in equal amounts, one with an electron-withdrawing functional group to protect the benzene ring from sulfonation, such as ketone or sulfone, and the other with a benzene ring structure that has a higher electron cloud distribution to provide a site for sulfonic acid group grafting. After copolymerization, an alternating (hydrophilic–hydrophobic) structure is formed. P4 has a molecular weight of 100,000 (g/mol), a glass transition temperature (Tg) of 332 °C, and an initial decomposition temperature (T_d5_%) of 562 °C, while T_d5_% is the temperature at which a sample has undergone 5% mass loss due to decomposition, desorption, or any other process causing a change in mass. It has a thermal expansion coefficient (TEC) of 19 ppm/°C, and a water absorption rate (WAR) of 0.1 and has the potential to be used as a flexible substrate for OLEDs [12,13,14]. The structure of P4 and some of its features are shown in Figure 1.

The experimental materials required for P4 polymerization are as follows: (1) 4,4′-(9-Fluorenylidene)diphenol (Aldrich, Cas: 3236-71-3, FW: 350.42); (2) 4,4″″-Difluoro-3,3″″-bistrifluoromethyl-2″,3″,5″,6″-tetraphenyl-[1,1′;4′,1″;4″,1‴;4‴,1″″]-pentaphenyl(FW: 858.86); (3) potassium carbonate (K_2_CO_3_) (Riedel-de Haen, CAS: 209-529-3, FW: 138.21, 99.8%); (4) toluene (ECHO, CAS: 67-56-1, FW: 32.04, 99.9%); (5) dimethylacetamide (DMAc) (Aldrich, CAS: 127-19-5, FW: 87.12, 99.8%); and (6) methanol (ECHO, CAS: 67-56-1, FW: 32.04, 99%). The experimental steps are as follows: Using a 100 mL three-necked round-bottomed reaction flask, 4,4′-(9-Fluorenylidene)diphenol (0.816 g, 2.32 mmol, 1 eq), 4,4″″-Difluoro-3,3″″-bistrifluoromethyl-2″,3″,5″,6″-tetraphenyl-[1,1′;4′,1″;4″,1‴;4‴,1″″]-pentaphenyl (2 g, 2.32 mmol, 1 eq), and potassium carbonate (0.71, 5.12 mmol, 2.2 eq) were added. A Dean–Stark trap was set up and covered with aluminum foil. Dimethylacetamide (25 mL) and toluene (15 mL) were added under a nitrogen atmosphere, and the mixture was heated to an internal temperature of 110 °C, after which a large amount of nitrogen gas was introduced. Water was removed from the reaction mixture by collecting the toluene using the azeotropic properties of toluene and water. After removing the toluene (16 mL), the aluminum foil around the reaction flask was removed, and the temperature was raised to 150–160 °C for 24 h. Samples were taken, and the molecular weight was tracked and detected using GPC. After the reaction was completed, the reaction mixture was cooled to room temperature, and a suitable amount of tetrahydrofuran was added to make the mixture viscous. Then, using a plastic dropper, a large amount of methanol was slowly added to the mixture to obtain white fibrous or granular precipitates. The material was then subjected to vacuum filtration and washed with DI water several times before being dried in an 80 °C oven. The dried material was dissolved in a suitable amount of tetrahydrofuran, and then slowly added to a large amount of methanol using a plastic dropper to obtain white fibrous and granular precipitates. The material was washed with DI water and subjected to vacuum filtration several times before being dried under vacuum at 80 °C for 12 h, resulting in 2.63 g of white fibrous polymer. PMMA (Mw = 350,000 g mol^−1^, Sigma Aldrich, Gillingham, UK) was dissolved in *N*,*N*-dimethylformamide (DMF, Sigma Aldrich, Gillingham, UK) to obtain a transparent polymer solution with a concentration of 12 wt%. The polymer was dissolved at 55 °C using the magnetic stirrer plate, set at a rotation speed of 700 rpm (IKA RCT basic, Staufen, Germany) for 2.5 h [15,16].

Polymer material P4 is dissolved in DMAC or NMP solvent in proportion to the required experimental amount to prepare a 10 wt% solution. To do this, the mixture is heated in an oven at 60 °C for one to two days to ensure complete dissolution. The solution is then transferred to a glove box for spin coating. Poly(methyl methacrylate) (PMMA) is supplied by Sigma-Aldrich (Mw = 350.000 g mol^−1^, Sigma Aldrich, Gillingham, UK). It is used as a surface flattening layer. PGMEA is chosen as the solvent for PMMA and is prepared into a 5 wt% solution according to the required experimental amount. The mixture is heated in an oven at 60 °C for one to two days before being transferred into a glove box for spin coating.

### 2.2. Spin Coating for Planarization Layer

We placed the glass in a staining jar and sequentially poured acetone, isopropanol (IPA), and deionized water (DI water) for cleaning [17,18]. We used an ultrasonic cleaner to clean the substrate surface for 10 min each and a lint-free cotton swab to wipe off any remaining particles. After confirming the cleanliness of the substrate surface, we used a nitrogen gun to blow off any remaining water droplets and finally placed it in an oven to bake at 100 °C for 10 min to remove any excess moisture.

For P4 (thickness of 3 μm) we used a calibrated dropper to take an appropriate amount of the P4 solution and drop it onto the cleaned glass surface. We used a two-step rotation parameter: 500 rpm for 5 s and 3000 rpm for 30 s. We completed the coating in a nitrogen-filled glove box and placed the sample on an electromagnetic heating plate, heating it at 100 °C for 30 min.

For P4 (thickness of 126 μm) we used a calibrated dropper to take an appropriate amount of the P4 solution and drop it onto the cleaned glass surface. We used a one-step rotation parameter at 100 rpm for 5 s. We completed the coating in a nitrogen-filled glove box and placed the sample on an electromagnetic heating plate, heating it at 60 °C for 5 h.

We used a calibrated dropper to take an appropriate amount of PMMA solution [19,20], and dropped it onto the substrate. We used a one-step rotation parameter of 1000 rpm for 30 s. We completed the coating in a nitrogen-filled glove box and placed the sample on an electromagnetic heating plate, heating it at 100 °C for 30 min. Finally, we transferred the sample into the vacuum chamber of the deposition machine and waited for the vacuum pumping step to be completed before proceeding with the deposition.

### 2.3. Preparation of Vapor-Deposited Materials

Anodized vapor deposition: We controlled the vacuum condition of the vapor deposition chamber to reach 1.2 × 10^−6^ torr before conducting vapor deposition. We preheated the ZnS for about 8 min during the vapor deposition process until the material was heated to the point of vaporization and sublimation. After pre-deposition of about 10 Å, we controlled the vapor deposition rate to maintain it at 0.2~0.4 Å/s, then vapor deposition was begun [21].

Before evaporating Ag, we preheated it for about 10 min. Once the material was heated to the point of vaporization and sublimation, we pre-deposited about 10 Å, and then controlled the evaporation rate to maintain it at 0.2~0.4 (Å/s) to begin the deposition.

Before evaporating MoO_3_ [22], we preheated it for about 10 min. Once the material was heated to the point of vaporization and sublimation, we pre-deposited about 10 Å, and then controlled the evaporation rate to maintain it at 0.1~0.2 (Å/s) to begin the deposition. Vapor deposition chamber for transmission layer: We preheated for approximately 10 min while vaporizing the NPB material, we then pre-deposited about 10 Å. Next, we maintained the deposition rate between 0.5 to 1.0 Å/s to commence the vapor deposition process. Vapor deposition of the emissive layer: We preheated for approximately 10 min while vaporizing the Alq3 material, then pre-deposited about 10 Å. We subsequently maintained the deposition rate between 0.5 to 1.0 Å/s to initiate the vapor deposition process. Vapor deposition of the electron injection layer: During the vaporization of LiF, we maintained the current at 60 amperes for 1 min, then increased it to 70 amperes for 3 min. Once the material had been heated to vaporization and sublimation, we controlled the deposition rate at 0.1 to 0.2 Å/s and commenced the vapor deposition. Vapor deposition of the cathode [23]: When vapor depositing the cathode aluminum (Al), we increased the current to 90 amperes for 1 min during preheating, then gradually raised it. As the current reaches approximately 100 amperes, the material begins to melt. We continued to increase the current to around 120 amperes, at which point the aluminum was heated to vaporization and sublimation. After a 15 Å pre-deposition, we maintained the deposition rate at 1.0 (nm/s) and initiated the vapor deposition process, achieving a thickness of approximately 1000 Å. The resulting high-polymer organic light-emitting diode components are depicted in Figure 2. Figure 2 shows P4 polymer organic light-emitting diode (OLED) devices with different area sizes. There are six OLED devices with an area size of 2 cm × 2 cm and 18 OLED devices with an area size of 1 cm × 1 cm. The fabrication process for each device includes the following: (1) cleaning the glass substrate; (2) spin-coating the P4 polymer; (3) spin-coating the PMMA; (4) depositing the anode layer by thermal evaporation; (5) depositing the hole transport layer by thermal evaporation; (6) depositing the emissive layer by thermal evaporation; (7) depositing the electron injection layer by thermal evaporation; and (8) depositing the cathode layer by thermal evaporation.

## 3. Results

### 3.1. Thin-Film Optical Property Analysis

Utilizing a UV-visible spectrophotometer, the transmission spectra of P4, PMMA, and PET thin films were measured, with respective thicknesses of 2.6 μm, 130 μm, and 0.3 μm. Measurements were taken under atmospheric pressure conditions, with a wavelength range of 200 nm to 800 nm. PMMA has a low refractive index, is colorless, and is transparent. After absorbing moisture, its heat distortion temperature decreases. PMMA demonstrates excellent stability against oxygen and ultraviolet (UV) radiation, resulting in high environmental tolerance. Flexible PET substrate was purchased from Wah Hong Industrial Corporation, with a film thickness of 130 µm. This polymer possesses advantages such as acid resistance, high transmittance, and low cost, making it widely used in the panel market. Figure 3 depicts the transmission spectra of P4, PMMA, and PET thin films [24,25,26]. The transmittances of P4, PMMA, and PET thin films at a wavelength of 550 nm are 96.9%, 97.3%, and 94.9%, respectively. These values exceed the general requirement of over 90% light transmittance at this wavelength for plastic substrates. This ensures that, after other processes, the plastic substrates achieve over 85% transmittance. Within the visible light range (400 nm to 800 nm), P4, PET, and PMMA have average visible light transmittance rates of 96.8%, 95.0%, and 97.4%, respectively.

Haze properties are measured using a UV-visible spectrophotometer. Haze refers to the percentage ratio of the scattered light flux to the transmitted light flux through a material, also known as turbidity. It is employed to gauge the degree of clarity or haziness in transparent or translucent materials and serves as an indicator of scattering. Haze results from light scattering caused by internal or external surface imperfections, creating a misty or cloudy appearance. A higher haze value implies reduced transparency of the thin film, softer emitted light, and stronger glare protection. The wavelength measurement range is between 200 nm and 800 nm, with the measurement conducted under atmospheric conditions. The material film formation and measurement conditions are identical to those of the transmission spectra. Haze measurement and calculation methods follow ISO 14782 [27].

Figure 4 presents the haze spectra of P4, PMMA, and PET thin films. The haze values of P4, PET, and PMMA at a wavelength of 550 nm are 18.7%, 20.4%, and 20.0%, respectively. Within the visible light range, P4, PET, and PMMA exhibit average haze values of 18.4%, 20.1%, and 19.7%, respectively.

### 3.2. Surface Morphology Analysis

#### 3.2.1. Analysis of Surface Morphology for Different Substrates

We fabricated P4 thin films of varying thicknesses at the fastest and slowest coating speeds, yielding measured thicknesses of 2.6 μm and 126 μm, respectively. As shown in Figure 5, to compare the changes in the surface roughness of P4 and PET before and after the addition of PMMA, we used the AFM technique. The AFM brand model used was a SEIKO SPA-300HV, and hardware control system specifications included the following: (1) scan speed: 0.05–125 Hz; nm~µm/s; (2) scan rotation angle: 360° (accuracy ±0.1°); (3) display chip resolution DSP: 40 bits; (4) bias on sample: ±10 V; and (5) control resolution: horizontal X-Y: 18 bits DAC ± 200 V; vertical Z: 21 bits DAC ± 200 V. Specifications for the 20 um scanner include the following: (1) scanning range: horizontal nm–20 µm; vertical −1.6 µm; (2) horizontal resolution ≤ 0.2 nm, vertical resolution ≤ 0.01 nm; and (3) manual mechanical movement range ±2.5 m. The mode employed was contact mode and the operating environment was an atmospheric vacuum. It was found that the root mean square (RMS) roughness of PET, P4 (2.6 μm), and P4 (126 μm) decreased after coating with PMMA, dropping from 1.1 nm, 8.6 nm, and 0.6 nm to 0.3 nm, 0.4 nm, and 0.3 nm, respectively. Moreover, all values were lower than the 1.3 nm roughness of commercially available glass substrates [28]. It is noteworthy that, as observed in Figure 6c,e, pore formation occurred in the pure P4 films of both thicknesses. However, after applying the PMMA planarization layer, these pores were filled. The holes generated on the surface of P4 can effectively be coated with PMMA as a surface planarization layer, resulting in a relatively smooth surface and improved hole injection capabilities after the application of PMMA.

Figure 6 and Table 1 show the comparative surface roughness of PET vs. PET/PMMA, P4 (2.6 μm) vs. P4 (2.6 μm)/PMMA, and P4 (126 μm) vs. P4 (126 μm)/PMMA thin films.

#### 3.2.2. Analysis of Anode Surface Morphology for Different Substrates

In this study, the ZnS/Ag/MoO_3_ system serves as a transparent conductive anode. The surface morphology and electrical properties of ZnS/Ag and ZnS/Ag/MoO_3_ are investigated. The metal layer primarily aims to achieve good conductivity and allow the multilayer film to attain high transmittance in the visible light spectrum and high reflectance in the infrared spectrum. However, Au, Ag, and Al all exhibit good conductivity. Ag demonstrates the lowest absorption rate (−5%) in visible light, whereas Au and Al exhibit absorption rates of 8% and 30%, respectively. Higher absorption rates reduce transmittance; thus, Ag was chosen as the metal layer. Semitransparent Ag (silver) thin films are often used in applications such as solar cells, touch screens, smart windows, and optical coatings due to their unique properties, which include high electrical conductivity and excellent optical performance. In the context of semitransparent Ag thin films, the color cast problem can arise due to the following reasons. Thickness variation: the thickness of the Ag thin film plays a crucial role in determining its optical properties. Variations in film thickness can lead to different colors appearing on the surface, which may be undesirable for certain applications. Surface roughness: the surface roughness of the Ag thin film can also contribute to the color cast issue. A rough surface scatters light in various directions, leading to unpredictable interference patterns and resulting in an unwanted color cast. Oxidation: silver is prone to oxidation, which can alter the optical properties of the film. The formation of silver oxide on the surface can cause a change in the film’s color, leading to an unwanted color cast. Several approaches can be used to address the issue of color cast in semitransparent Ag thin films: optimizing thickness, surface treatment and applying a protective layer, can all help to prevent oxidation and reduce the impact of plasmonic effects. In conclusion, while semitransparent Ag thin films offer many advantages, the problem of color cast can be a significant drawback. By understanding the underlying causes and exploring various approaches to mitigate this issue, it is possible to improve the performance and appearance of devices utilizing these films.

To mitigate reflection caused by the metal, a high refractive index dielectric layer is necessary. ZnS (n = 2.3) possesses a high n value, making it suitable for use as an antireflection layer [29]. Figure 7 presents the three-dimensional surface morphology of ZnS (30 nm)/Ag (15 nm) deposited on different substrates, while Figure 8 shows the surface roughness for the same deposition. The root-mean-square roughness (RMS) decreases from 3.4 nm to 1.6 nm. Similarly, for glass substrates, the RMS declines from 2.8 nm to 1.3 nm after PMMA application. On the other hand, the roughness of PET and P4 (126 μm) increases from 1.2 nm and 1.5 nm to 1.5 nm and 1.7 nm, respectively. Commercially available ITO films exhibit a surface roughness of 2.3 nm. This study reveals that applying PMMA to different substrate-based anodes results in smoother surfaces and improved hole injection capabilities [30]. Table 2 compares the surface roughness after depositing ZnS (30 nm)/Ag (15 nm) on different substrates.

### 3.3. Analysis of Anode Thin Film Resistance

A four-point probe was used to analyze the resistivity of the thin film. Because Ag is a good conductor, the ZnS buffer layer shows very high resistance. If the measured thin film resistance is very low, the value should be considered as the resistance of the Ag thin film alone, indicating a continuous Ag film. If the measured thin film resistance is higher, the value should be regarded as the resistance of the substrate or the buffer layer, and that the silver film is discontinuous [29]. The thin film resistance of PET, P4 (2.6 μm), and P4 (126 μm) on different substrates for depositing ZnS (30 nm)/Ag (15 nm) and ZnS (30 nm)/Ag (15 nm)/MoO_3_ (5 nm) slightly decreased after adding PMMA. In the case of ZnS/Ag, the values decreased from 6.88 Ω/sq, 8.61 Ω/sq, and 6.07 Ω/sq to 5.75 Ω/sq, 6.66 Ω/sq, and 5.98 Ω/sq, respectively, and as shown in Figure 9 and Figure 10. However, the thin film resistance of the glass substrate did not decrease after adding PMMA, instead increasing from 5.12 Ω/sq to 5.84 Ω/sq. Nevertheless, the anode thin film resistances mentioned above are all lower than the value of commercially available ITO, which is about 14 Ω/sq. This thin layer resistance is inversely proportional to the carrier mobility and conductivity, and the factors affecting the multilayer film conductivity include the thickness and surface roughness of the Ag layer [31]. It can be seen that, due to the thickness of the evaporated MoO_3_ being 5 nm, the thin film resistance measured for ZnS/Ag/MoO_3_ does not increase much compared with ZnS/Ag, so MoO_3_ will not have a significant impact on the charge balance and electrical properties of OLEDs [32].

### 3.4. Optical Simulation Analysis

In order to understand the impact of the filled holes in P4, Rsoft 8.0 simulation software was used to perform finite-difference time-domain (FDTD) optical simulations. The structures of P4 and PET were compared, and the refractive index profile of the P4 structure corresponding to the surface morphology measured by previous AFM are shown in Figure 11. As can be seen from Figure 12, the calculation results for the 550nm green light wavelength reveal that, compared with PET components, P4 components significantly increase the chances of extracting light, with the electromagnetic field distribution in the P4 substrate being stronger than that in PET.

From Figure 13, it can be observed that the radiant flux measured by the software detectors for both components initially increases rapidly and then reaches a stable saturation state. The energy detected in the P4 component is relatively higher than that in the PET component. This is mainly due to the particle diameter of the P4 substrate being around 0.3 μm. This microstructure size is close to the 550 nm green light wavelength, which can provide additional wave vectors for some confined waves (guided light modes, transverse surface plasmon resonance modes). Due to the conservation of momentum, confined photons are transformed into escapable photons, enhancing the light extraction phenomenon. Moreover, the holes are not created through a particularly complex process, but simply by coating and drying the P4 material.

Figure 14 shows the results of optical simulations of the P4 component structure at 480 nm blue light wavelength and 620 nm red light wavelength, yielding similar increased light extraction results as at 550 nm wavelength. This suggests that this microstructure is suitable for other colored OLED components and has potential applications in displays and lighting.

### 3.5. Device Performance Analysis

Figure 15 illustrates the structure of the prepared devices, differing only in the nature of the substrate used. Figure 16 displays the luminance–voltage characteristics of different devices. It is observed that the maximum luminance of PET device, after incorporating a PMMA planarization layer, increases two-fold from 4300 cd/m² to 8800 cd/m². This is attributed to the decrease in film resistance after adding PMMA, despite the surface roughness of the anode being unimproved. Likewise, the luminance of the PET, P4 (126 μm), and P4 (2.6 μm) devices increases significantly, and the turn-on voltage is reduced when PMMA is added.

Figure 17 presents the power efficiency–voltage characteristics of different devices. After adding PMMA, the power efficiency of PET, P4 (126 μm), and P4 (2.6 μm) devices improves significantly, with the highest efficiency observed for P4 (2.6 μm) devices. Figure 18 displays the current efficiency–voltage characteristics for different devices, revealing that the current efficiency of PET, P4 (126 μm), and P4 (2.6 μm) devices increases significantly after adding PMMA, with P4 (2.6 μm) devices showing the highest current efficiency.

Figure 19 depicts the relationship between external quantum efficiency (EQE) and voltage for different devices. The EQE of PET, P4 (126 μm), and P4 (2.6 μm) devices is significantly higher after adding PMMA, with the highest EQE observed for P4 (2.6 μm) devices. The relevant data for different devices are summarized in Table 3.

From the comprehensive perspective above, the luminance, power efficiency, current efficiency, and external quantum efficiency of P4 (2.6 μm) devices are the highest after adding PMMA. The optimal flexible organic light-emitting diode device substrate material is chosen to be P4 (2.6 μm) with PMMA. Further optimization of the emitting layer thickness is carried out with P4 (2.6 μm)/PMMA as the substrate material. It is well known that the quantum efficiency of OLEDs is affected by the balance of electrons and holes, as well as the position and density of excitons in OLEDs. Efficiency improvement can be achieved by controlling charge balance within the device [33].

Due to the naturally formed pores in P4 films on the substrate, the reproducibility, brightness, and efficiency of devices were investigated. Three different emitting layer thicknesses—50 nm, 45 nm, and 55 nm—were used for emitting layer thickness optimization. It was anticipated that electron hole recombination will be more efficient, and that the reproducibility of devices with a P4 film thickness of 2.6 μm would be confirmed. As shown in the electroluminescent spectra of Figure 20, all three devices with different thicknesses remain in the green light wavelength range. There is a slight redshift of the peak with increasing thickness, and the brightness is relatively close. The maximum luminance of 45 nm, 50 nm, and 55 nm devices is 86,000 cd/m², 87,900 cd/m², and 93,400 cd/m², respectively. The relevant parameters of the P4 (2.6 mm) devices are listed in Table 4.

## 4. Conclusions

The maximum brightness of flexible organic light-emitting diodes (FOLEDs) depends on various manufacturing techniques and materials. According to research reports, there are already some high-brightness FOLED products on the market, with brightness ranging from 1000 to 10,000 cd/m^2^ per square meter, which is sufficient for indoor and outdoor lighting. One of the brightest flexible organic light-emitting diode products currently available on the market is “P-OLED” developed by South Korean company LG Display, with a maximum brightness of around 1000 nits (cd/m²). In this study, we successfully used our self-developed material, polyaromatic ether polymer P4, as a substrate for producing flexible organic light-emitting diodes. To confirm the reproducibility of naturally formed pores, we made the components again with three different emission layer thicknesses (Alq3). When the Alq3 thickness was 55 nm, the maximum brightness of our components reached 93,400 cd/m^2^, with a maximum external quantum efficiency of 1.7% and a maximum current efficiency of 5.6 cd/A, which is far greater than the maximum brightness of current flexible organic light-emitting diode products on the market. From the surface morphology and film resistance of the thin film, we found that, when the P4 thickness was 2.6 μm, the roughness was relatively large due to the generation of micron-sized pores but that this was reduced by the PMMA flatting layer, thereby reducing the film resistance. Furthermore, this particular microstructure can improve light extraction efficiency, leading to better component efficiency and characteristics.

## 5. Patents

Patents resulting from the work reported in this manuscript: US8,987,407B2; US9,748,594B2; US9,644,069B2; US9,209,472B2; US9,018,336B2.

## Figures and Tables

**Figure 1 micromachines-14-00920-f001:**
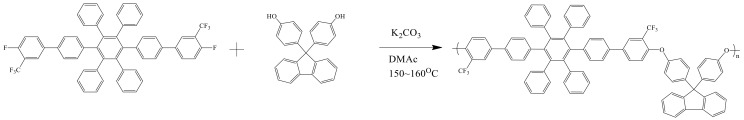
P4 polymer structural formula.

**Figure 2 micromachines-14-00920-f002:**
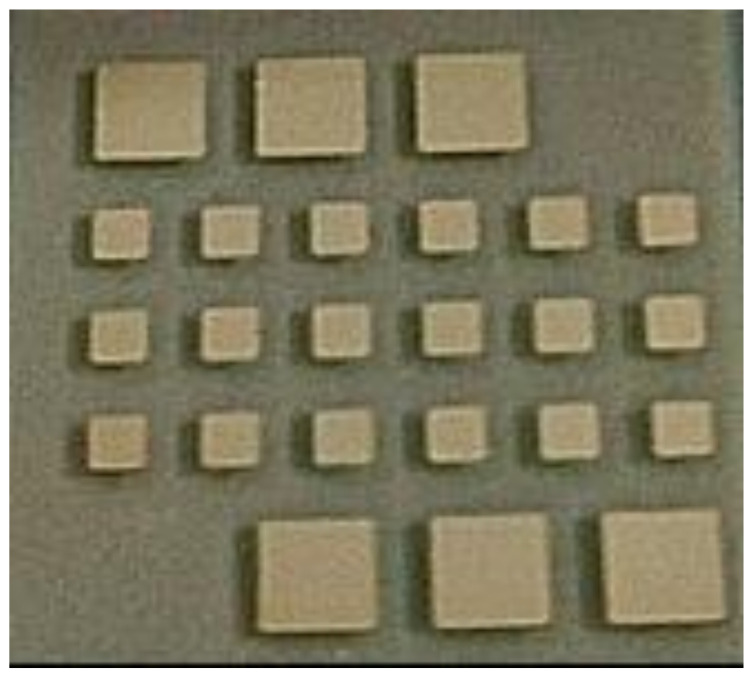
The polymer organic light-emitting diode.

**Figure 3 micromachines-14-00920-f003:**
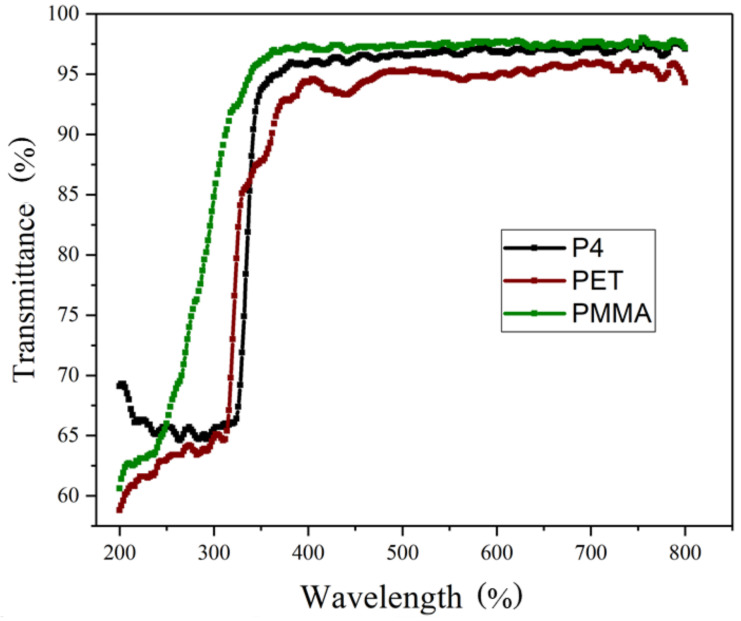
Transmission spectra of P4, PMMA, and PET thin films.

**Figure 4 micromachines-14-00920-f004:**
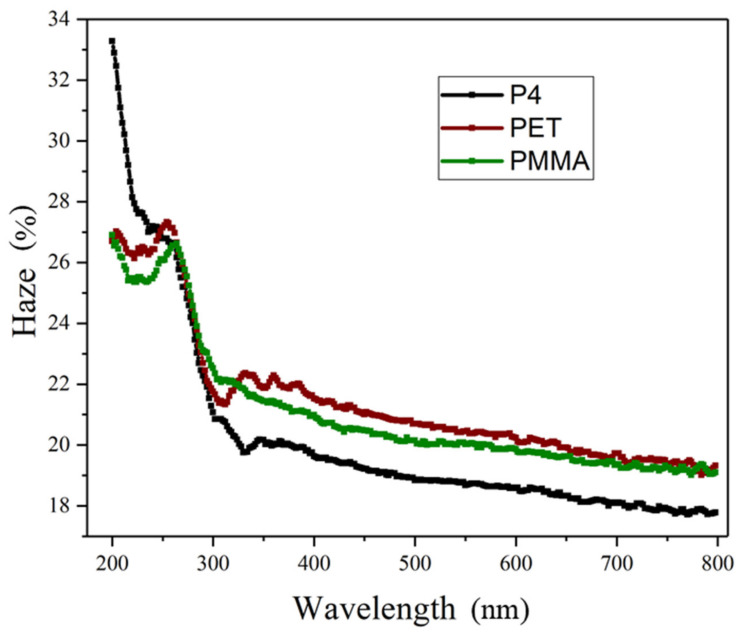
Haze spectra of P4, PMMA, and PET thin films.

**Figure 5 micromachines-14-00920-f005:**
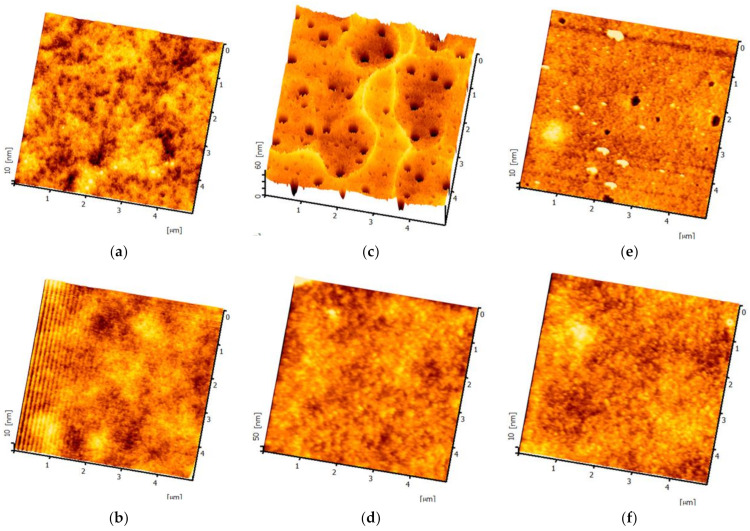
Three-dimensional surface morphology of thin films: (**a**) PET, (**b**) PET/PMMA, (**c**) P4 (2.6 μm), (**d**) P4 (2.6 μm)/PMMA, (**e**) P4 (126 μm), and (**f**) P4 (126 μm)/PMMA.

**Figure 6 micromachines-14-00920-f006:**
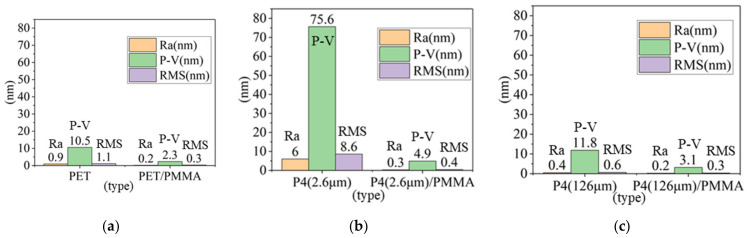
Surface roughness comparison of (**a**) PET vs. PET/PMMA, (**b**) P4 (2.6 μm) vs. P4 (2.6 μm)/PMMA, and (**c**) P4 (126 μm) vs. P4 (126 μm)/PMMA.

**Figure 7 micromachines-14-00920-f007:**
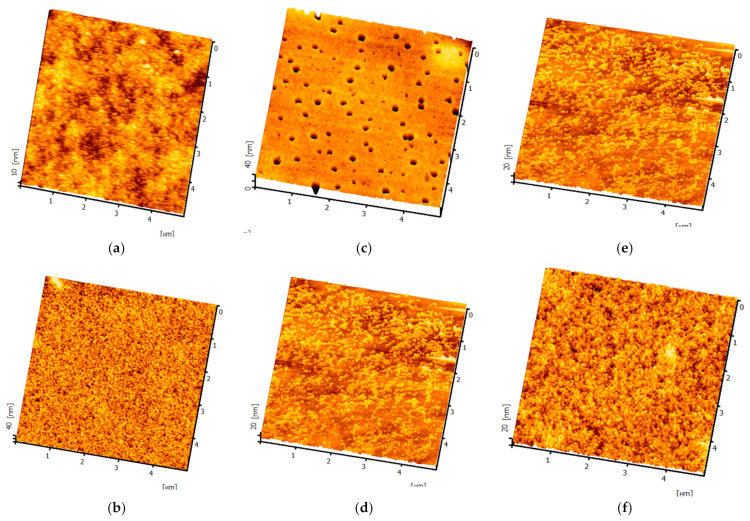
Three-dimensional surface morphology of ZnS/Ag deposited on different substrates: (**a**) PET, (**b**) PET/PMMA, (**c**) P4 (2.6 μm), (**d**) P4 (2.6 μm)/PMMA, (**e**) P4 (126 μm), (**f**) P4 (126 μm)/PMMA.

**Figure 8 micromachines-14-00920-f008:**
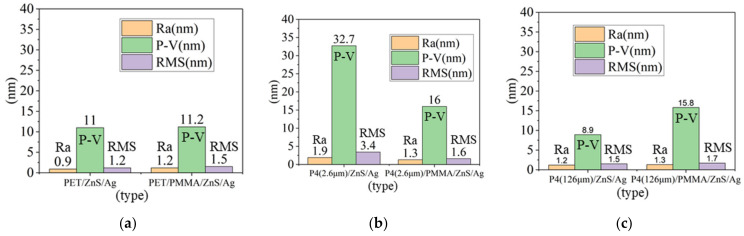
Surface roughness of ZnS/Ag deposited on (**a**) PET and PET/PMMA substrates, (**b**) P4(2.6 μm) and P4 (2.6 μm)/PMMA substrates, and (**c**) P4 (126 μm) and P4 (126 μm)/PMMA substrates.

**Figure 9 micromachines-14-00920-f009:**
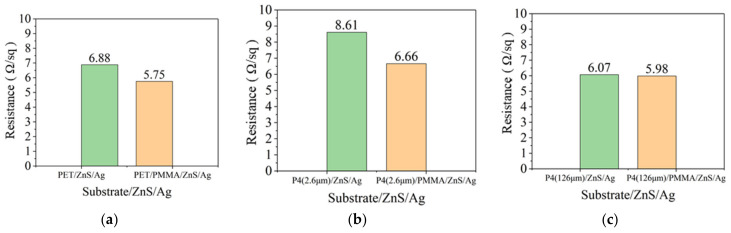
(**a**) Thin film resistance of PET and PET/PMMA. (**b**) Thin film resistance of P4 (2.6 μm) and P4 (2.6 μm)/PMMA. (**c**) Thin film resistance of P4 (126 μm) and P4 (126 μm)/PMMA.

**Figure 10 micromachines-14-00920-f010:**
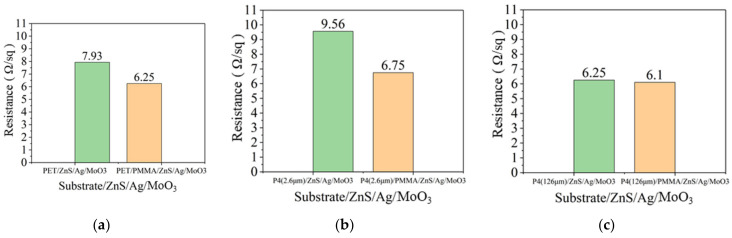
(**a**) Thin film resistance of PET and PET/PMMA substrates with ZnS/Ag/MoO_3_ deposition (**b**). Thin film resistance of P4 (2.6 μm) and P4 (2.6 μm)/PMMA substrates with ZnS/Ag/MoO_3_ deposition. (**c**) Thin film resistance of P4 (126 μm) and P4 (126 μm)/PMMA substrates with ZnS/Ag/MoO_3_ deposition.

**Figure 11 micromachines-14-00920-f011:**
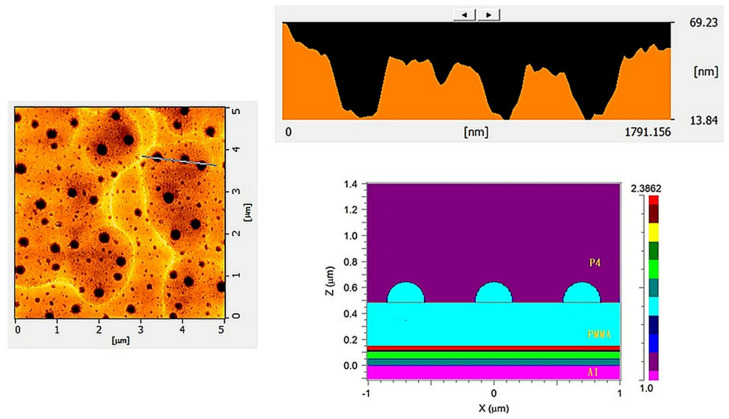
The optical simulation calculation of the refractive index profile for the P4 structure.

**Figure 12 micromachines-14-00920-f012:**
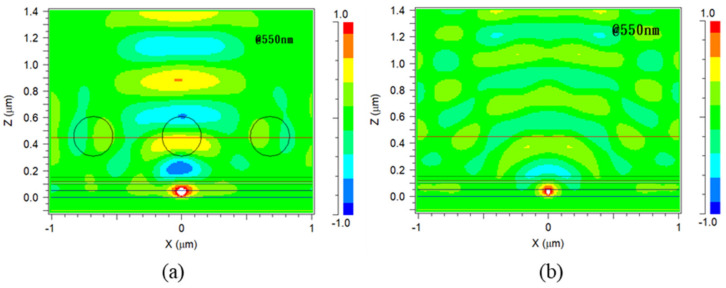
(**a**) The light field distribution of P4 components. (**b**) The light field distribution of PET components.

**Figure 13 micromachines-14-00920-f013:**
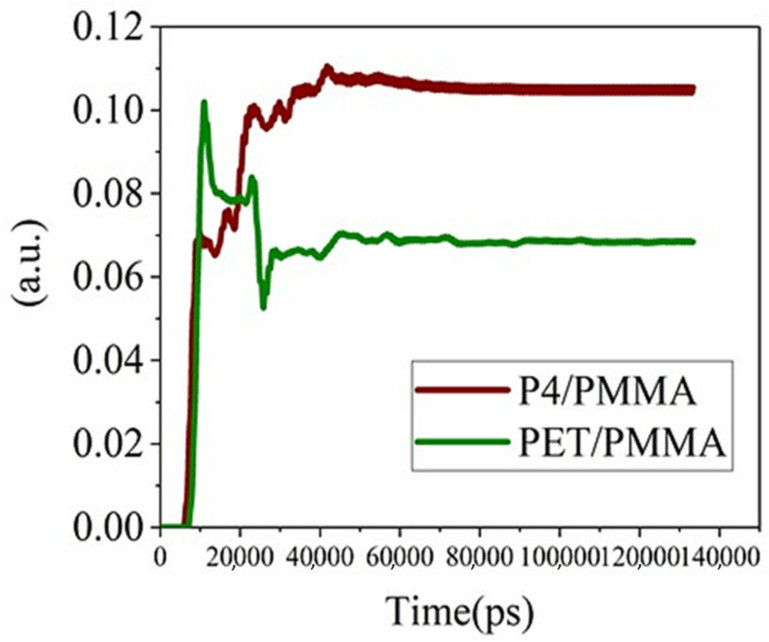
The radiation flux versus time curves for P4 and PET.

**Figure 14 micromachines-14-00920-f014:**
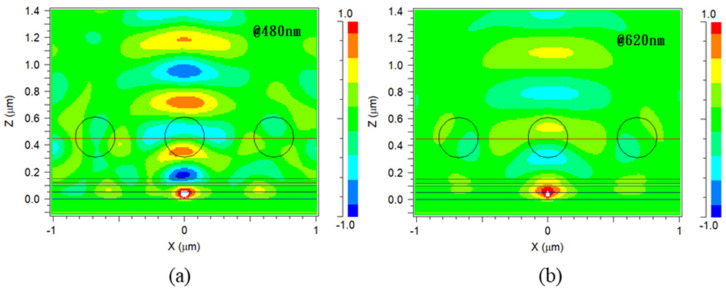
(**a**) The light field distribution of P4 components at 480 nm wavelength. (**b**) The light field distribution of P4 components at 620 nm wavelength.

**Figure 15 micromachines-14-00920-f015:**
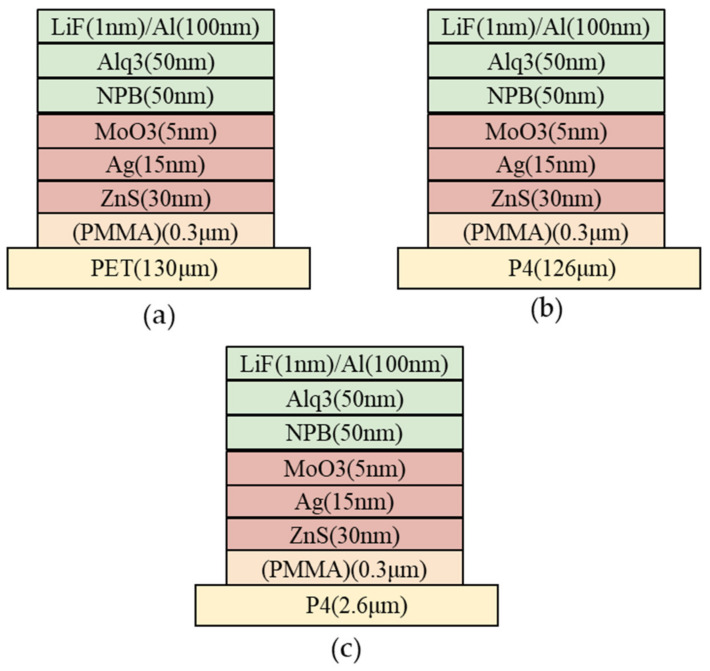
Device structures based on different substrates; (**a**) PET (130 mm), (**b**) P4 (126 µm), and (**c**) P4 (2.6 µm).

**Figure 16 micromachines-14-00920-f016:**
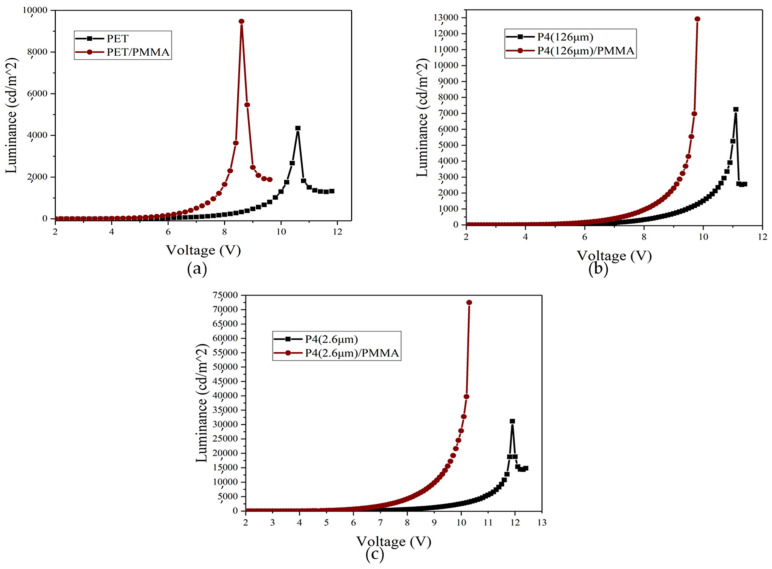
Luminance–voltage curves of different devices with and without incorporation of PMMA; (**a**) PET (130 mm) device, (**b**) P4 (126 µm) device, and (**c**) P4 (2.6 µm) device.

**Figure 17 micromachines-14-00920-f017:**
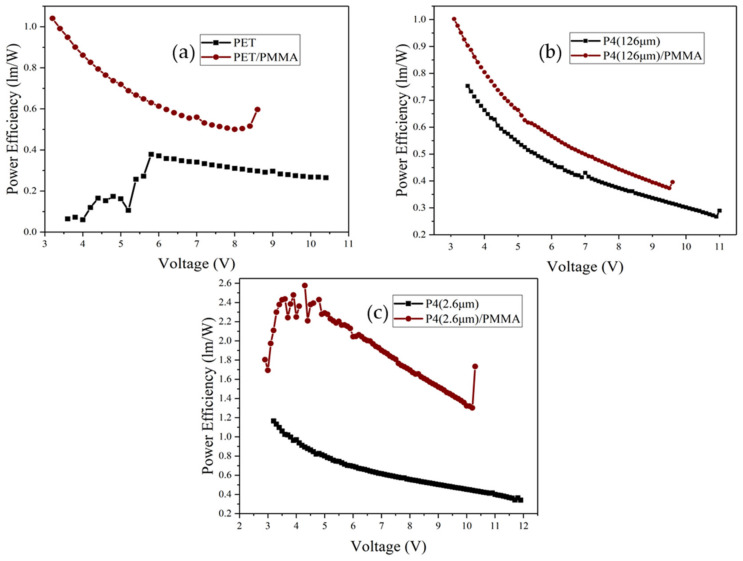
Power efficiency–voltage curves of different devices with and without incorporation of PMMA; (**a**) PET (130 mm) device, (**b**) P4 (126 µm) device, and (**c**) P4 (2.6 µm) device.

**Figure 18 micromachines-14-00920-f018:**
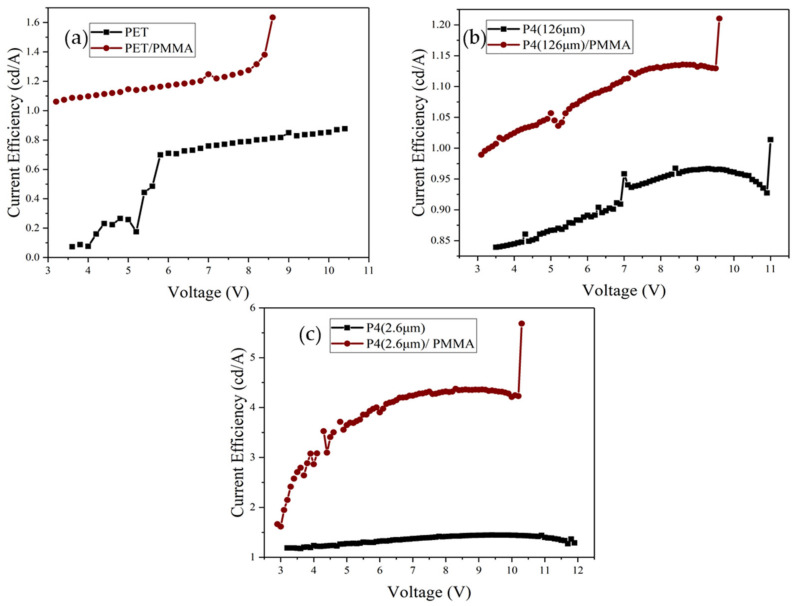
Current efficiency–voltage curves of different devices with and without incorporation of PMMA; (**a**) PET (130 mm) device, (**b**) P4 (126 µm) device, and (**c**) P4 (2.6 µm) device.

**Figure 19 micromachines-14-00920-f019:**
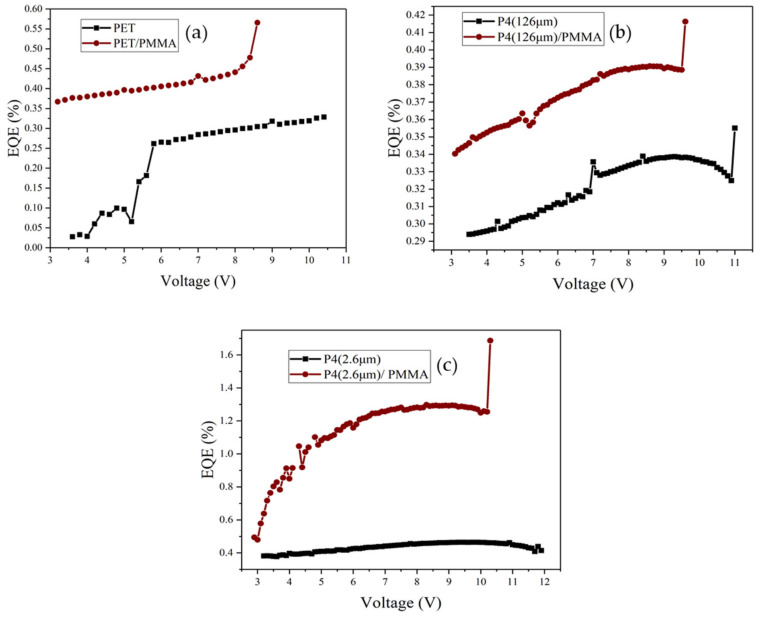
External quantum efficiency (EQE) vs. voltage curves for different devices with and without incorporation of PMMA; (**a**) PET (130 mm) device, (**b**) P4 (126 µm) device, and (**c**) P4 (2.6 µm) device.

**Figure 20 micromachines-14-00920-f020:**
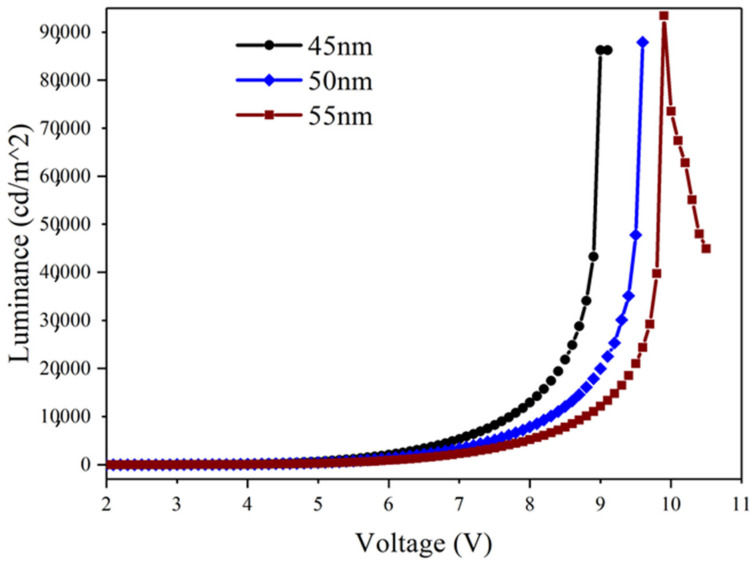
Relationship between different emission layer thicknesses (Alq3) and voltage.

**Table 1 micromachines-14-00920-t001:** Comparison of thin film surface roughness.

Materials	Ra (nm)	P-V (nm)	RMS (nm)
PET	0.9	10.5	1.1
PET/PMMA	0.2	2.3	0.3
P4 (2.6 μm)	6	75.6	8.6
P4 (2.6 μm)/PMMA	0.3	4.9	0.4
P4 (126 μm)	0.4	11.8	0.6
P4 (126 μm)/PMMA	0.2	3.1	0.3

**Table 2 micromachines-14-00920-t002:** Comparison of surface roughness for ZnS (30 nm)/Ag (15 nm) deposited on different substrates.

Materials	Ra (nm)	P-V (nm)	RMS (nm)
PET	0.9	11	1.2
PET/PMMA	1.2	11.2	1.5
P4 (2.6 μm)	1.9	32.7	3.4
P4 (2.6 μm)/PMMA	1.3	16	1.6
P4 (126 μm)	1.2	8.9	1.5
P4 (126 μm)/PMMA	1.3	15.8	1.7

**Table 3 micromachines-14-00920-t003:** Relevant data for different devices.

Materials	The Initial Voltage (Von, V)	cd/A (@1000 cd/m^2^)	Maximum Current Efficiency (cd/A)	EQE Max (%)	Maximum Power Efficiency (lm/W)	Lmax (cd/m^2^)
PET (130 μm)	3.6	0.8	0.9	0.3	0.4	4300
PET (130 μm) /PMMA (0.3 μm)	3.2	1.2	1.17	0.6	0.93	8800
P4 (126 μm)	3.5	1.0	1	0.4	0.8	7200
P4 (126 μm) /PMMA (0.3 μm)	3.1	1.1	1.36	0.42	1	12,900
P4 (2.6 μm)	3.2	1.4	1.4	0.5	1.2	31,100
P4 (2.6 μm) /PMMA (0.3 μm)	2.9	4.1	5.68	1.69	2.58	72,500

**Table 4 micromachines-14-00920-t004:** Device parameters for different emission layer thicknesses (Alq3).

Alq3 Thickness	The Initial Voltage (Von, V)	cd/A (@1000 cd/m^2^)	Maximum Current Efficiency (cd/A)	EQEmax (%)	Maximum Power Efficiency (lm/W)	Lmax (cd/m^2^)
45 nm	2.6	3.7	4.9	1.4	3.85	86,000
50 nm	2.7	3.5	4.2	1.3	1.97	87,900
55 nm	2.7	3.5	5.6	1.7	3.4	93,400

## Data Availability

Not applicable.

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
