# Peer review of "Application of Self-Assembled Polyarylether Substrate in Flexible Organic Light-Emitting Diodes"

_micromachines, 2023, doi:10.3390/mi14050920_

Round 1

Reviewer 1 Report

The authors repot a polyarylether substrate for making flexible organic light-emitting diodes. The topic is interesting, because flexible organic light-emitting diodes are of wide use. I can recommend the manuscript for the publication, after the authors address the following issues.

1.       The author should elucidate the mechanism of the creation of holes on the P4 surface. In addition, the image of the cross section of P4 film is needed, in order to see the depth of these holes.

2.       The optical simulation is missing from the main text. The supplementary figure and table are not mentioned in the main text.

3.       The photostability of P4, PET, and PMMA should be compared.

4.       The semitransparent Ag thin film causes a severe problem of color cast. The authors should discuss it.

5.       The flexible organic light emitting diodes must be discussed enough in the introduction.

6.       There are a number of expression errors. For example,

In the abstract, P4 should be replaced by its full name.

In the table captions, the abbreviations like Von should be defined.

“which is far greater than the maximum brightness of current flexible organic light-emitting diode products on the market.” is not proper.

7.       The caption of figure 3 should be revised.

The major revisions are needed.

   The entire manuscript is redundant and should be simplified.

Reviewer 2 Report

The authors present a well-organized study on flexible OLEDs prepared with a unique architecture using several flexible polymeric materials as substrate. The devices were also characterized after adding PMMA and for different emission layer thickness. In this sense, the authors develop a good presentation of the techniques used and their results; however, some aspects are still needed in the discussion section. Questions and corrections to the text are pending.

Based on the novelty of the study and the weight of the evidence presented and discussed, I recommend that the manuscript be returned to the author for major revision and thus be accepted for publication.

 Questions and correction (line)

 Page 1

Replace “MoO3” by “MoO3” (13, 14); “480nm, 550nm, and 620nm” by “480 nm, 550 nm, and 620 nm” (19); “2.6mm” and “5.68cd/A” by “2.6 mm” and “5.68 cd/A”, respectively (21); “55nm” and “5.6nm” by “55 nm” and “5.6 nm”, respectively (29); “10V” by “10 V” (45).

Page 2

Replace “P4 has a glass…” by “P4 has a molecular weight of 100,000 (g/mol), a glass…” (90), “materials” by “Materials” (83), “332°C” by “332 °C” (90), “562°C” and “19ppm/°C” by “562 °C” and “19 ppm/°C”, respectively (91) and “P4 is a material developed in our laboratory and used as the substrate for flexible components in the experiment. Its structure is shown in Figure 1 and is an alternating copolymer with a molecular weight of 100,000 (g/mol). Its substrate characteristics are shown in Table 1” by “The structure of P4 and some of its features are shown in Figure 1 and Table 1, respectively.” (94-97).

Page 3

Replace “After confirming the preparation, the solution is heated in an oven at 60°C…” by “To do this, the mixture is heated in an oven at 60 °C…” (103-104), “PMMA stands for Poly(methyl methacrylate) and is supplied by Sigma-Aldrich” by “Poly(methyl methacrylate) (PMMA) is supplied by Sigma-Aldrich (incorporate here the information of molecular weight and polydispersity index) (105-106), “After preparation, the solution” by “The mixture”, and “100°C” by “100 °C” (118).

Remove “as shown in Figure 2” and modify the number of the figures to come (107), Remove Figure 2 and its legend (111-112).

Remove Table 1 (its content was given in the text (90-92)). Despite its elimination, doubts are raised about the definition of the term Td5 (%). It is usually the temperature at which 5% of the initial mass is lost in a TGA analysis, however, in the text it is defined as the decomposition start temperature (90), which is Tonset; They are different parameters. This should be made clear in the text. On the other hand, the Tg of the material is valued at 332 °C in the text (90), while the table shows 331 °C. Although the difference is negligible, it should be clarified which is the correct value.

Page 4

Replace “Take out the dissolved P4 solution and place the cleaned glass on the rotating table of a spin coater. Use a calibrated dropper to take an appropriate amount of solution and drop it evenly onto the glass surface” by “Use a calibrated dropper to take an appropriate amount of the P4 solution and drop it onto the cleaned glass surface” (120-122) and (126-128), “100°C” by “100 °C” (125, 135), “60°C” by “60 °C” (130), “Take out the dissolved PMMA solution [17-18], remove the sample, place it on the rotating table of a spin coater, use a calibrated dropper to take an appropriate amount of PMMA solution, and drop it evenly onto the substrate” by “Use a calibrated dropper to take an appropriate amount of PMMA solution [17-18], and drop it onto the substrate”(130-133), “1.2x10^(-6)” by “1.2x10-6” (140), “0.2~0.4Å/s” by “0.2~0.4 Å/s” (143), “MoO3” by “MoO3” (147), “1.0” by “1.0 (incorporate here units)” (164).

Page 5

Figure 3 does not contribute to the understanding of the text since it does not specify in detail which are the components that are being displayed. If this is not specified in a new version of the manuscript, I strongly recommend its removal.

Replace “optical property analysis” by “Optical Property Analysis” (170), “2.6µm, 130µm, and 0.3µm” by “2.6 µm, 130 µm, and 0.3 µm” (172-173), “200nm to 800nm. Polymethyl methacrylate (PMMA)” by “200 nm to 800 nm. PMMA” (174), “Polyethylene terephthalate (PET)” by “Flexible PET substrate was purchased from Wah Hong Industrial Corporation, with a film thickness of 130 µm (incorporate information from the molecular weight and polydispersity index). This polymer” (177), “550nm” by “550 nm” (181), “a 550nm” by “this” (182), “400nm to 800nm),” by “400 nm to 800 nm),” (184).

Page 6

In Figure 4 replace “Transmittance(%)” and “Wavelength(nm)” by “Transmittance (%)” and “Wavelength (nm)”, respectively.

Replace “200nm and 800nm” by “200 nm and 800 nm” (196), “550nm” by “550 nm” (200).

Remove “(400nm to 800nm)” (201).

Page 7

In Figure 7 replace “Haze(%)” and “Wavelength(nm)” by “Haze (%)” and “Wavelength (nm)”, respectively.

Replace “morphology analysis” by “Morphology Analysis” (205), “2.6µm and 126µm.” by “2.6 µm and 126 µm” (208), “As shown in Figure 6, comparing the changes in P4 and PET before and after the addition of PMMA, we employed AFM to measure the surface roughness of P4 and PET” by “As shown in Figure 6, to compare the changes in the surface roughness of P4 and PET before and after the addition of PMMA, we used the AFM technique” (210-211), “2.6µm”, “126µm”, “1.1nm”, “8.6nm”, “0.6nm”, “0.3nm”, “0.4nm”, “0.3nm”, and “1.3nm” by “2.6 µm”, “126 µm”, “1.1 nm”, “8.6 nm”, “0.6 nm”, “0.3 nm”, “0.4 nm”, “0.3 nm”, and 1.3 nm”, respectively (212-214), “(c)” by “(6c)”, “P4 films” by “pure P4 films” (216), “2.6µm” and “126µm” by “2.6 µm” and “126 µm”, respectively (218-219).

Remove “The flexible PET substrate was purchased from Wah Hong Industrial Corporation, with a PET film thickness of 130µm” (208-209).

It remains to comment on what happened with the PET sample before and after treatment with PMMA (AFM results).

Page 8

Figure 6. In order to make comparisons easier, it is suggested to place the photographs in pairs (a)-(b); (c)-(d) and (e)-(f) (220-223).

Replace “2.6µm” and “126µm” by “2.6 µm” and “126 µm”, respectively (224-225, 228-229).

In Figure 7 It is necessary to indicate the name of the X-axis, since only the unit of measurement is provided. Ra and P-V are not defined in the text. Neither does the manuscript present a discussion that uses the parameters shown in the figure and in Table 2 (230).

Page 9

Table 2: Replace “Ra(nm)”, “P-V(nm)”, and “RMS(nm)” by “Ra (nm)”, “P-V (nm)”, and “RMS (nm)”, respectively (230), “P4(2.6µm)” and “P4(126µm)” by “P4 (2.6 µm)” and “P4 (126 µm)”, respectively (230), “P4(2.62.6µm)” by “P4 (2.6 µm)”.

Incorporate reference for “The surface morphology and electrical properties of ZnS/Ag and ZnS/Ag/MoO3 are investigated”.

Replace “n=2.3”, “30nm”, “15nm”, “P4(2,6mm)”, “3,4nm”, “1.6nm”, “2.8nm”, “1.3nm”, “P4(126mm)”, “1.2nm”, “1.5nm”, “1.7nm”, “2.3nm” by “n = 2.3”, “30 nm”, “15 nm”, “P4 (2,6 mm)”, “3,4 nm”, “1.6 nm”, “2.8 nm”, “1.3 nm”, “P4 (126 mm)”, “1.2 nm”, “1.5 nm”, “1.7 nm”, “2.3 nm” (between 240-252),

The idea “It is evident that P4(2.6µm) initially exhibits noticeable pores before the application of PMMA, which are improved after the PMMA coating” does not correspond to what is discussed. This effect was analyzed in (212-214). As I understand it, Figures 9c and 9e show the surface of Zn/Ag deposition on pure PET of different thicknesses, preserving the morphology of the substrate. When the deposit is made on PET-PMMA (9d and 9f), the surface roughness decreases notably, reproducing the morphology of the substrate used.

Figure 8. In order to make comparisons easier, it is suggested to place the photographs in pairs (a)-(b); (c)-(d) and (e)-(f) (253-256).

Page 10

Replace “P4(2.6µm)” and “P4(126µm)” by “P4 (2.6 µm)” and “P4 (126 µm)”, respectively (258, 262).

In Table 3, replace “Materials” by “Substrates”, “Ra(nm)”, “P-V(nm)”, and “RMS(nm)” by “Ra (nm)”, “P-V (nm)”, and “RMS (nm)”, respectively, “P4(2.6µm)”, “P4(126µm)”, and “P4(2.62.6µm)” by “P4 (2.6 µm)”, “P4 (126 µm)”, and “P4 (2.62 µm)” respectively, “anode thin film resistance” by “Anode Thin Film Resistance” (265), “By using a four-point probe to analyze the thin film resistivity, and because Ag is a good conductor, the ZnS buffer layer has a very high resistance” by “A four-point probe was used to analyze the resistivity of the thin film. Because Ag is a good conductor, the ZnS buffer layer shows very high resistance”, “As shown in Figures 10 and Figures 11, the thin…” by “The thin…” (271), “P4(2.6µm)” and “P4(126µm)” by “P4 (2.6 µm)” and “P4 (126 µm)”, respectively (272).

Page 11

Replace “ZnS(30nm)/Ag(15nm) and ZnS(30nm)/Ag(15nm)/MoO3(5nm) slightly…” by “ZnS(30 nm)/Ag(15 nm) (Figure 10) and ZnS(30 nm)/Ag(15 nm)/MoO3(5 nm) (figure 11), slightly…” (273), “14w/sq” by “14 w/sq” (278), “This thin…” by “The thin…” (278).

In X-axes of Figure 10 and Figure 11 replace “(w/sq)” by “Resistance (w/sq)”, In Y-axes of Figure 11 replace “MoO3” by “MoO3”.

Replace “P4(2.6µm)” and “P4(126µm)” by “P4 (2.6 µm)” and “P4 (126 µm)”, respectively (292-293), “Figure 12 illustrates the structure of PET devices, P4(126µm) devices, and P4(2.6µm) devices” by “Figure 12 illustrates the structure of the prepared devices, differing only in the nature of the substrate used”, “devices” by “device” (298), “As shown in Figure 13, the luminance of PET, P4(126µm)…” by “Likewise”, the luminance of P4 (126 mm)…” (301).

Page 12

Replace “PET”, “P4(2.6µm)”, and “P4(126µm)” by “PET (130 mm)”, “P4 (2.6 µm)”, and “P4 (126 µm)”, respectively (between 302-318), “devices” by “device” (315), “50nm, 45nm, and 55nm” by “50 nm, 45 nm, and 55 nm” (324), “emission layer thicknesses…” by “emission layer thicknesses (Alq3)…”, and “P4 devices” by “P4 (2.6 mm) devices” (331).

Remove “(OLED)” (316).

There is no explanation for the increase in power efficiency-voltage when PMMA is incorporated in the different devices (306), the same is true for current efficiency and external quantum efficiency results (309, 313, respectively).

The sentence “…and the reproducibility of devices with a P4 film thickness of 2.6 µm will be confirmed…” seems to be not so true, since the maximum luminance value shown in Figure 13c (~73000 cd/m2) and in Figure 17 (87900 cd/m2) do not coincide despite the fact that the same Alq3 thickness (50 nm) is used. An explanation should be given for this fact.

Page 13

Modify Figure 12, leaving only one structure where the base is labeled "Substrate".

Replace “Structures of different devices (a) Structure of PET device (b) Structure of P4(126µm) device (c) Structure of P4(2.6µm) device” by “Device structures based on different substrates; (a) PET (130 mm), P4 (126 µm), and (c) P4 (2.6 µm)” (333-334).

Page 14

Replace “…different devices (a) Luminance-voltage curve of PET device (b) Luminance-voltage curve of P4(126µm) device (c) Luminance-voltage curve of P4(2.6µm)” by “…different devices with and without incorporation of PMMA; (a) PET (130 mm) device (b) P4 (126 µm) device, and (c) P4 (2.6 µm) device (336-338), “…different devices (a) Power efficiency-voltage curve of PET device (b) Power efficiency-voltage curve of P4(126µm) device (c) Power efficiency-voltage curve of P4(2.6µm) device” by “…different devices with and without incorporation of PMMA; (a) PET (130 mm) device (b) P4 (126 µm) device, and (c) P4 (2.6 µm) device” (340-342).

Page 15

Replace “…different devices (a) Current efficiency-voltage curve of PET device (b) Current efficiency-voltage curve of P4(126µm) device (c) Current efficiency-voltage curve of P4(2.6µm) device” by “…different devices with and without incorporation of PMMA; (a) PET (130 mm) device (b) P4 (126 µm) device, and (c) P4 (2.6 µm) device” (344-346).

Page 16

Replace “…different devices (a) EQE-voltage curve of PET device (b) EQE-voltage curve of P4(126µm) device (c) EQE-voltage curve of P4(2.6µm) device” by “…different devices with and without incorporation of PMMA; (a) PET (130 mm) device (b) P4 (126 µm) device, and (c) P4 (2.6 µm) device” (344-346), “layer thicknesses and voltage” by “layer thicknesses (Alq3) and voltage” (354), “Device parameters for different emission layer thicknesses” by “P4 (2.6 mm)/PMMA device parameters for different emission layer thicknesses” (355).

In Table 5 replace “45nm”, “50nm”, and “55nm” by “45 nm”, “50 nm”, and “55 nm”, respectively.

Page 17

Replace “55nm”, “5.6cd/A”, and “2.6mm” by “55 nm”, “5.6 cd/A”, and “2.6 mm”, respectively (between 367-372), “thismanuscript” by “this manuscript” (377),

It is not possible to access Supplementary Materials, so this report was not evaluated.

 Referee

Corrections to the writing format are necessary, as well as in the wording of some important ideas.

Reviewer 3 Report

         I recommend the publication of the manuscript after a minor revision.

Line 3: Delete the dot from the end of the title.

        Line 10: mistake: only one author with  “† These authors contributed equally to this work.” If you specify it, it must be a minimum of 2 co-authors.

Line 167: can you specify all components from Fig. 3.

Line 205: Can you specify these microtexture parameters for the samples: a) Skewness Ssk [-]; Kurtosis Sku [-]?

Line 211: Please, provide the names of the used equipment and tools.  For example for AFM: “3-D surface topography was recorded using a Nanoscope Multimode atomic force microscope (Digital Instruments, Santa Barbara, CA), in tapping mode and a scan speed of 10-20 μm/s to obtain 256 × 256-pixel images.  The experiments were carried out at room temperature (297 ± 1 K), using cantilevers with the following nominal properties for force-distance curve measurements: length 180 µm, width 25 µm, thickness 4 µm, tip radius 10 nm, quality factor Q = 100, mass density ρ = 2330 kg/m3, Young's modulus E = 1.3 x 1011 Pa, and Poisson ratio ν = 0.28” and so on. “ 

Line 334: improve the resolution of Fig. 13.

Line 336: improve the resolution of Fig. 14.

Line 343: improve the resolution of Fig. 15.

Line 347: improve the resolution of Fig. 16.

Line 377: minor mistake “.....in thismanuscript:.......”

Line 379-380: delete or fill with the corresponding data.

Line 387: delete the dot “.....Taiwan. and...”.

Line 387: delete the symbol “.....-086-.”

Lines 108-110: insert more details and explanations.

Insert a paragraph with Statistical analyses, and explain the method, the software used, and all the parameters related to these statistical experiments.

Please, state the manufacturer's name, city, and country from where the equipment was sourced.

Specify the limits of this study.

I recommend inserting References:

https://doi.org/10.1016/j.xcrp.2021.100408, https://doi.org/10.1002/app.49763

This paper can be published after the mentioned revisions.

Minor editing of English language required.

Round 2

Reviewer 1 Report

The authors revise their manuscript accroding to the reviewer's comments. I can recommend it for publicaiton. 

The authors are advised to improve the English level of their manuscript with the aid of the native English speaker.